# Classification of periodontitis stage and grade using natural language processing techniques

**Nazila Ameli** [iD][1]*, **Tahereh Firoozi**[1], **Monica Gibson**[2], **Hollis Lai**[1]

**1** Mike Petryk School of Dentistry, Faculty of Medicine and Dentistry, University of Alberta, Edmonton, Alberta, Canada, **2** Department of Periodontology, School of Dentistry, University of Indiana, Indianapolis, United States of America

* nazila@ualberta.ca

**Data Availability Statement:** The data that support the findings of this study are available from the University of Alberta but restrictions apply to the availability of these data, which were used under

## Abstract

Periodontitis is a complex and microbiome-related inflammatory condition impacting dental supporting tissues. Emphasizing the potential of Clinical Decision Support Systems (CDSS), this study aims to facilitate early diagnosis of periodontitis by extracting patients' information collected as dental charts and notes. We developed a CDSS to predict the stage and grade of periodontitis using natural language processing (NLP) techniques including bidirectional encoder representation for transformers (BERT). We compared the performance of BERT with that of a baseline feature-engineered model. A secondary data analysis was conducted using 309 anonymized patient periodontal charts and corresponding clinician's notes obtained from the university periodontal clinic. After data preprocessing, we added a classification layer on top of the pre-trained BERT model to classify the clinical notes into their corresponding stage and grades. Then, we fine-tuned the pre-trained BERT model on 70% of our data. The performance of the model was evaluated on 32 unseen new patients' clinical notes. The results were compared with the output of a baseline feature-engineered algorithm coupled with MLP techniques to classify the stage and grade of periodontitis. Our proposed BERT model predicted the patients' stage and grade with 77% and 75% accuracy, respectively. MLP model showed that the accuracy of correct classification of stage and grade of the periodontitis on a set of 32 new unseen data was 59.4% and 62.5%, respectively. The BERT model could predict the periodontitis stage and grade on the same new dataset with higher accuracy (66% and 72%, respectively). The utilization of BERT in this context represents a groundbreaking application in dentistry, particularly in CDSS. Our BERT model outperformed baseline models, even with reduced information, promising efficient review of patient notes. This integration of advanced NLP techniques with CDSS frameworks holds potential for timely interventions, preventing complications and reducing healthcare costs.

## Author summary

In our study, we aimed to enhance the early diagnosis of periodontitis, a complex inflammatory dental condition, by developing a Clinical Decision Support Systems (CDSS)

license for the current study, and so are not publicly available. Data are however available at dentrsch@ualberta.ca upon reasonable request and with permission of University of Alberta.

**Funding:** This work was supported by the Network for Canadian Oral Health Research (NCOHR) New Frontier Seed Grant Program (2020–2021), awarded to HL, NA, and MG, and a graduate student scholarship from Alberta Innovates (2023–2024), awarded to NA. The funders had no role in the study design, data collection and analysis, decision to publish, or preparation of the manuscript.

**Competing interests:** The authors have declared that no competing interests exist.

through the application of natural language processing (NLP) techniques. Specifically, we developed a CDSS utilizing bidirectional encoder representation for transformers (BERT) to predict the stage and grade of periodontitis by analyzing patients' dental charts and notes. We conducted a secondary data analysis using anonymized patient records from a university periodontal clinic, comparing the performance of our BERT model with a baseline feature-engineered model. Our results demonstrated that our BERT model achieved higher accuracy in predicting both the stage and grade of periodontitis compared to the baseline model. Specifically, the BERT model predicted the stage and grade with 77% and 75% accuracy, respectively, outperforming the baseline model's accuracy of 59.4% and 62.5%. This novel application of BERT in dentistry, particularly in CDSS, holds promise for a more efficient review of patient notes, enabling timely interventions to prevent complications and reduce healthcare costs.

## Introduction

Periodontitis is a multifactorial and microbiome-associated inflammatory disease that occurs in the dental supporting tissues [1,2]. Progression of the disease can adversely affect oral and systemic health and result in tooth loss, reduction of masticatory performance [3] as well as having association with diabetes [4], and rheumatoid arthritis [5]. Thus, periodontitis and its complications will impose substantially negative effects on oral health related quality of life (OHRQoL), while successful and timely diagnosis and management of the disease may improve patients' OHRQoL [3,6]. Moreover, early detection and diagnosis of periodontitis can help in preventing the consequent costly and invasive dental treatment [2].

According to the 2017 World Workshop on the classification of Periodontal and Peri-implant Diseases and Conditions [1], the recent method for classifying periodontitis is based on staging and grading. Staging is determined through the severity of the disease and complexity of its management, while grade is an indicator of the rate of periodontitis progression assessed according to the history and the presence of risk factors for the disease. According to the criteria defined for stage and grade of the periodontal disease [1], clinicians traditionally analyze patients' systemic, clinical, and radiographic data collected in periodontal charts over time to determine the stage and grade of periodontitis. In recent decades, electronic dental records (EDR) have been introduced for collection of patient data, which are found to have superiority over paper charts in terms of higher storage capacity, time efficiency, ease of information retrieval and accessibility [7]. In addition, new computerized techniques known as artificial intelligence (AI) and its subsets like machine learning (ML) provide opportunities to extract valuable information from complex data and analyze various relationships to benefit patients [8]. AI is an evolving field that seeks to automate tasks that would require intelligence if it was done by humans. ML also has the potential to monitor and detect patterns of patient presentations and risk factors [9,10].

Recently, neural networks (NN) models- a powerful tool in ML- are gaining attraction and are being extensively used in the area of diagnosis [11], prognosis [12], classification [13], and predictions [14,15] due to their ability of modelling non-linear relationships among the hidden variables in different formats of data including images and texts. Moreover, advancement in text mining and natural language processing (NLP) techniques—also a branch of AI techniques- such as Bidirectional Encoder Representation for Transformers (BERT), significantly improved the accuracy of text classification systems using deep learning models (DL) [16]. NLP methods and text mining approaches have been widely used for extracting information

from the patient's electronic records in medicine and dentistry [17–20]. Chen et al., applied NLP techniques by implementing Sentence2vec and Word2vec approaches to learn sentence vectors and word vectors, and to extract information from Chinese EDR. They reported that their NLP workflow can efficiently structure narrative text from EDR [21]. Patel et al. in a recent study developed and applied two automated NLP algorithms (approximate string-matching function and Levestein Distant Function) to extract information from clinical notes to track periodontal disease change over time using longitudinal EDR. They concluded that utilizing longitudinal EDR data to track disease changes over 15 years was a feasible method that could be applied for studying clinical courses using AI and ML methods [22].

BERT is a language model which uses transformers for text representation, and supports the pretraining and fine tuning of the model in the case of textual data. Application of BERT and related architectures have resulted in considerable improvements in multiple medical applications, including processing of electronic health records [23], outcome prediction [24], identification of medical terms and concepts [25], and others. Despite its success in the medical field, there are no studies that assess the BERT application in predicting the stage and grade of the periodontal disease using patients' textual notes. Regular expression (RegEx) is another powerful tool for pattern matching and text analysis. By defining a sequence of characters in a string using literal text or special characters with specific meaning, the tool is able to search, match, and manipulate text data.

It has been shown that AI techniques such as text mining and DL algorithms can be successfully applied for extracting semantic information from narrative patients' charts in dentistry, classifying the periodontal disease and predicting the factors influencing the periodontitis occurrence [8,26]. Compared to the rule-based approaches using NLP tools, DL models do not depend on the grammatical accuracy of sentences, so can extract implicit aspects following their identification [27]. However, the use of AI and DL techniques to review narrative patients' charts and determine the contribution of its information to the stage and grade of periodontitis in addition to clinical and radiographic findings has not yet been studied to the authors' knowledge. Thus, the aim of the present study is to introduce and compare automated methods to:

1. Extracting crucial latent information from patients' unstructured notes and chart data, employing a baseline rule-based model (RegEx),

2. Employing the extracted information to classify the stage and grade of periodontitis using an MLP algorithm,

3. Automating the classification of patients with periodontitis based on unstructured patient notes, utilizing BERT and classification models.

4. Comparing and contrasting the result of pretrained models including BERT with that of rule-based models Regex.

This research aims to pioneer innovative approaches within dentistry, particularly in leveraging CDSS, to enhance early diagnosis and treatment of periodontitis.

## Methods

To demonstrate the applicability of the proposed method for accurate and timely classification of the stage and grade of the periodontitis using the possible contributing factors mentioned in clinicians' notes and periodontal charts, we conducted a secondary data analysis on the intake patient charts and textual notes referred to a University Periodontal Graduate Clinic from 2017 to 2021. Since our research predominantly relied on secondary data analysis, obtaining

patient consent was not relevant. The data were anonymized and the University Human Research Ethics Board approved this study (Pro00107743). Three hundred and nine patient charts and relevant notes were collected in the initial visit.

The proposed method consists of two phases of AI algorithms to facilitate information extraction and classification. In the initial phase, the state-of-the-art BERT model was employed for comprehensive text analysis and classification of the periodontitis stage and grade. The application of transformers in NLP tasks was accelerated with the advent of a specific transformer-based language model called the Bidirectional Encoder Representation for Transformers or BERT [28]. BERT is a transformer-based encoder model for language representation that uses a multi-head attention mechanism and a bidirectional approach to learn the contextual relations between words and sentences in a text for an accurate representation of the entire text. [29].

For training the BERT model, three hundred and nine patient clinical notes belonging to 309 patients referred to the University Periodontal Graduate Clinic were selected among the pool of 1513 patients' charts. These clinical notes met the inclusion criteria, which involved having the stage and grade of the disease determined by a graduate student and confirmed by a periodontist. The prevalence of different periodontitis stage and grades in the collected notes were as follows: stage I = 6, stage II = 36, stage III = 206, stage IV = 61, and grade A = 17, grade B = 202, grade C = 87.

We first imported the 309 textual patients' notes coupled with their stage and grade levels into the Google Colaboratory and preprocessed the texts for computational analysis. The data preprocessing stage involved meticulous steps to prepare the textual patient notes for analysis. This included tokenization, addition of special tokens such as [CLS] and [SEP], and strategies like padding and truncation to ensure uniformity in the input data format [30]. Furthermore, the dataset was split into training (80%) and testing (20%) subsets to evaluate model performance effectively.

After preprocessing the data, the necessary libraries including the "class_weight" module from scikit-learn [31], which aids in addressing class imbalance issues in classification tasks, were imported. The predefined window length in our study was 100. This number is determined based on the distribution of essay length in the proposed dataset. The model architecture employed combines elements of BERT and LSTM (Long Short-Term Memory) layers to address the classification task. BERT, a pre-trained language model, serves as the foundation for understanding contextual information within textual data [29]. Specifically, the "bert-base-uncased" model was utilized, known for its effectiveness in various NLP tasks.

The input sequences were passed through the BERT model, and a recurrent network was used to aggregate the multiple embeddings into a single embedding after the transformer window swipes over the long text. The LSTM model was used as the recurrent layer to aggregate the BERT output at each time step. This model combines the data into a sequence of vectors having the same length relative to their temporal position and temporal dependency with respect to the features in the essays [32]. LSTM was selected over other potential operations because the LSTM layers tend to produce more accurate modelling results of deep connections between sequential features that can be used to improve score prediction for text classification tasks [32]. By combining BERT's contextual understanding with LSTM's ability to model sequential data, the model aimed to capture both local and global dependencies within the input data, enhancing its predictive capabilities.

The output of the LSTM layer was subsequently passed through additional dense and dropout layers with ReLU activation functions. These layers enabled the model to learn higher-level features and prevent overfitting by introducing regularization. The final layer consisted of a fully connected neural network layer with softmax activation, producing classification

probabilities for each class. For each input text, the selected text stage or text grade was the class with the highest probability in the neural network's output. The number of units in this layer corresponds to the number of unique classes present in the training data (4 stages and 3 grades).

For model compilation, the Adam optimizer was employed with a predefined learning rate of 5e-6. The choice of sparse categorical cross-entropy as the loss function was appropriate for multi-class classification tasks, facilitating the comparison between predicted and actual class labels. During model training, accuracy was used as the evaluation metric to provide insight into the model's performance across different epochs.

To address potential class imbalance issues, class weights were computed and incorporated into the training process. These weights ensure that the model assigns appropriate importance to each class during optimization, thereby mitigating the impact of imbalanced data distributions. Additionally, early stopping was implemented as a regularization technique to prevent overfitting based on validation loss. Through monitoring the validation loss, the early stopping mechanism terminates training when the model's performance on unseen data begins to deteriorate. By restoring the best weights observed during training, this approach helps to optimize the model's generalization ability and improve its performance on unseen data.

The model was trained over 40 epochs with a batch size of 20 for both stage and grade and the generalizability of the model was evaluated by testing on a set of 32 new unseen patients' charts (stage I = 1, stage II = 4, stage III = 21, and stage IV = 6, grade A = 2, grade B = 21, and grade C = 9) using the evaluation metrics. Table 1 represents the model architecture and selected hyperparameters for training.

In the second phase, we developed a baseline rule-based model by employing traditional text mining and pattern extraction techniques followed by the utilization of an MLP model, for predictive classification. This phase involved mining textual data and extracting pertinent patterns to facilitate the MLP in discerning and categorizing the data accurately. Clinicians' notes, along with periodontal charts containing essential clinical and radiographic findings, served as primary data sources.

Text mining is a computerized technique to extract key information from vast quantities of textual data. It can be used for information retrieval, information extraction, and text categorization as a powerful research tool [33]. In the field of dentistry, text mining has been shown to be a valuable method of extracting latent (unknown) patterns from patient charts [15]. To complete our text mining, patient clinical notes were first imported and compiled into Google

**Table 1. Architecture and hyperparameters for the BERT model.**

| Layer | Parameter Name | Candidate Values | Selected Value |
|---|---|---|---|
| BERT | Number of parameters | 110M | 110M |
| | Transformer blocks | 12 layers | 12 layers |
| | Attention heads | 12 | 12 |
| | Hidden neurons | 768 | 768 |
| Dropout | Dropout rate | 0.2–0.5 | 0.2 |
| LSTM | Decay Rate | 0.96–0.97 | 0.97 |
| | Activation Function | ReLU, sigmoid | ReLU |
| | Learning Rate | 0.1-10e-7 | 5e-6 |
| | Momentum | 0.5–0.9 | 0.5 |
| Dense | Neurons | 25–100 | 64 |
| Model Compile | Epoch | 15–50 | 40 |
| | Batch Size | 12–36 | 20 |

Colaboratory as individual text files. Then, a data-frame was made containing two columns: patients' chart numbers and clinicians' notes as unstructured texts. Upon conducting an initial assessment of the document, specifically with regard to the structure of the conformance rules text, we employed regular expressions (RegEx) to exclusively extract the unformatted text of those rules and the dependent variables (stage/grade), and possible contributing factors (including the patients' medical and dental history), and to identify and locate the pertinent patterns [34]. With RegEx search, particular strings of characters can be done using pattern matching, which is in contrast to constructing multiple, literal search queries [35]. To make the RegEx insensitive to the lowercase/capital letters, we used the command "re.IGNORE-CASE" following the definition of the pattern of each variable.

After finding the pattern for all possible variables embedded in clinicians' notes, we created a new data-frame with 16 columns including: patient chart number, date, stage, grade, systolic blood pressure, diastolic blood pressure, heart rate, tooth stain, smoking history, plaque, calculus, bone loss, tooth mobility, allergies, history of previous periodontal surgery, and diabetes. During this phase, one limitation we encountered was that not all records provided comprehensive information about patients' clinical and dental history including details such as history of previous periodontal surgery, smoking history, and amount of stain on the teeth. To address this, we implemented the RegEx extractor aiming for maximum information detail. However, if certain structured pattern were missing, the RegEx would not generate an error but instead extract the available limited information from the clinical note. For instance, if the clinical note did not mention previous periodontal surgery GBR/GTR/FGG, the program would still extract this information in a structured format, leaving the category of previous surgery empty.

The output data-frame was cleaned before exporting as a CSV file to facilitate future ML analysis. Table 2, shows the variables and their descriptive values (mean ± SD, minimum and maximum, or codes used for variables with categorical scale).

Finally, patients' periodontal charts including PPD, CAL, and the number of teeth with bleeding and/or plaque in CSV format were also imported into Google Colaboratory. Data cleaning and recoding were completed as explained above and the newly cleaned and re-coded periodontal charts were combined with the clinician's notes into one data-frame and exported as a CSV file (Table 3).

To classify the stage and grade of the periodontitis using contributing variables (15 extracted variables from clinicians' notes and 4 variables from periodontal charts), we first imported the CSV file created through the previous steps into Google Colaboratory. First, the dependent (stage or grade) and independent variables were defined as y and x, respectively. Then, the dataset was split into training (70%) and testing (30%) sets to train the network for classifying the target variables. To design our MLP architecture, we defined the grid with two hidden layers and varying numbers of neuron units within each hidden layer (ranging from 4–20 nodes). We empirically concluded that for classifying both the grade and stage of the periodontal disease, by increasing the number of hidden layers beyond two, the algorithm performance in terms of accuracy decreases. Finally, for this multi-class classification problem, a softmax function was applied in the output layer to yield the probability of each class at each unit of the output layer. The Rectifier Linear Unit (ReLU) was used in all four hidden layers because models working with ReLUs are more easily optimized compared to networks with sigmoid or tanh units [36]. The grid output revealed the optimal number of nodes in each hidden layer for classifying the target variables as follows: 15 and 4 nodes in the first hidden layer and 13 and 4 nodes in the second hidden layer for classifying stage and grade, respectively.

For training a NN, there are several optimization algorithms to choose from. Optimizers are algorithms or methods used to change the attributes of the NN such as weights and

**Table 2. Descriptives of extracted variables from the textual notes and periodontal charts.**

| Variable | Value (min-max, mean ± SD)/ Recoded values |
|---|---|
| Stage | I = 1, II = 2, III = 3, IV = 4 |
| Grade | A = 1, B = 2, C = 3 |
| Heart rate | 44–118, 72.4 ± 10.45 |
| Blood pressure (Systolic and diastolic blood pressure) | 79–187, 129.3 ± 15.54<br>43–120, 78.1 ± 40.32 |
| History of periodontal surgery (FGG[a], GTR[b], GBR[c]) | No = 0, Yes = 1 |
| Tooth stain | No = 0, Light = 1, Medium = 2, Heavy = 3 |
| Calculus | No = 0, Light = 1, Medium = 2, Heavy = 3 |
| Smoking history | No = 0, Yes = 1 |
| Bone loss | No = 0, Yes = 1 |
| Tooth mobility | No = 0, Yes = 1 |
| Diabetes | No = 0, Yes = 1 |
| Allergies | No = 0, Yes = 1 |
| Count of teeth with pockets | 8–32 |
| Pocket score | Localized = 0, Generalized = 1 |
| Count of teeth with CAL[d] | 8–32 |
| CAL score | Localized = 0, Generalized = 1 |

[a] free gingival graft

[b] guided tissue regeneration

[c] guided bone regeneration

[d] clinical attachment loss

**Table 3. Example of the collected CSV data file extracted from the patients' charts and clinical notes.**

| Variable | Value | | |
|---|---|---|---|
| Chart Number | 82 | 93 | 104 |
| Date | 2019-10-18 | 2019-06-17 | 2019-06-17 |
| Systolic blood pressure | 132 | 144 | 134 |
| Diastolic blood pressure | 78 | 88 | 86 |
| Stage | III | IV | III |
| Grade | A | C | B |
| Heart rate | 72 | 68 | 82 |
| Stain | Medium | Heavy | Heavy |
| Plaque | Heavy | Heavy | Medium |
| Calculus | Heavy | Heavy | Medium |
| Bone loss | Generalized | Generalized | Generalized |
| Tooth mobility | Yes | Yes | No |
| Allergy | No | No | No |
| History of previous surgery | No | Yes | No |
| Diabetes | No | Yes | No |
| Smoking | No | Yes | No |
| Pocket score | Generalized | Generalized | Generalized |
| CAL score | Generalized | Generalized | Generalized |
| Count of teeth with bleeding | 12 | 23 | 14 |
| Count of teeth with plaque | 22 | 17 | 19 |

*CAL: clinical attachment loss, FGG: free gingival graft, GTR: guided tissue regeneration, GBR: guided bone regeneration

learning rate in order to reduce the losses and increase the accuracy of the model [37]. One of the most popular is Adam, known for its efficiency and training speed [38]. To compile the model, we also used the "categorical_crossentropy" loss function. Finally, we fitted our model with the batch size of 16 and 30 epochs for both stage and grade outcomes. The evaluation metrics employed to present and interpret the results.

## Evaluation of model performance

Evaluation metrics included accuracy, which is the ratio of correctly predicted observations to the total observations; recall, which is the ratio of true positives to the sum of true positives and false negatives; precision, which is the ratio of true positives to the sum of true positives and false positives; and the F1-score, which is the harmonic mean of precision and recall [39]. These metrics range from 0 to 1, with 1 indicating perfect performance [40].

## Results

Our proposed BERT model for predicting the periodontitis stage and grade resulted in a high accuracy. The model predicted the patients' stage with 77% accuracy. Although the accuracy of the BERT model for patients' grades is high (accuracy = 75%), the performance of the model for predicting the stage of the patients' is higher than the grade.

The model performed quite accurately in classifying stage III and grade B, while in other stages and grades, the model couldn't perform accurately. The precision column indicated that our model was successful in assigning 91% and 65% of the patients in stage III and grade B to the right class, respectively. The recall column showed that 75% of the patients that truly belonged to stage III were identified correctly by our model; however, these values were found to be 47% and 35% for grades B and C, respectively. The F1-Score column, which is usually used to judge the overall performance of the model and is defined as the average of precision and recall values in the previous columns, showed that our proposed model performed highly accurately in predicting the patients with stage III (82%) compared to the periodontitis grade (Table 4).

We implemented model interpretability techniques such as Local Interpretable Model-Agnostic Explanations (LIME) to analyze the predictions made by the BERT model. LIME helps in identifying the key features that influence the model's decision-making process for classifying periodontitis stages and grades. This interpretability approach allows us to better understand the reasoning behind correct and incorrect classifications, providing insights into the strengths and weaknesses of the model. However, it is important to note that the primary aim of our study was not to conduct a detailed analysis of the specific features driving the predictions but to demonstrate the overall classification accuracy and applicability of these models

**Table 4. Confusion matrix evaluating the performance of the BERT model in classifying the stage and grade of the periodontal disease.**

| Stages/grades | Precision | Recall | F-Score |
| --- | --- | --- | --- |
| I | 0.11 | 0.25 | 0.15 |
| II | 0.25 | 0.30 | 0.27 |
| III | 0.91 | 0.75 | 0.82 |
| IV | 0.06 | 0.25 | 0.09 |
| A | 0.05 | 0.00 | 0.00 |
| B | 0.65 | 0.47 | 0.55 |
| C | 0.16 | 0.35 | 0.22 |

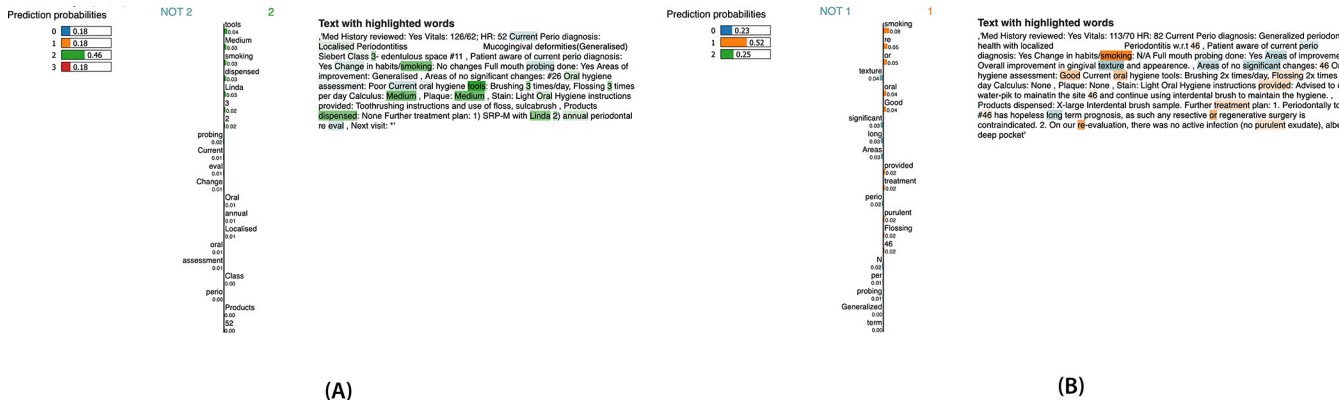

**(A)**  **(B)**

**Fig 1. LIME-based interpretation of BERT model predictions.** (A) Key features for classifying stage III vs. non-stage III. (B) Significant words and features for distinguishing grade B vs. non-grade B.

for periodontal staging and grading. Fig 1 (A and B) illustrates the most influential words and features contributing to the model's predictions for stage III and grade B as an example.

Table 5 depicts a full list of the features extracted from the notes and charts using the RegEx. MLP analysis demonstrated that the model can correctly classify the stage and grade of periodontitis by 69.9% and 69.8% accuracy, respectively using the extracted variables from patients' notes and periodontal charts. The highest accuracy was found to be for correct classification of stage III and grade B periodontitis (74.7% and 71.1%, respectively).

The performance of the model is shown in Table 6.

Table 7 compares the predictability of BERT and MLP models for periodontitis staging and grading on the set of new 32 data (stage I = 1, stage II = 4, stage III = 21, and stage IV = 6,

**Table 5. Example of the collected CSV data file extracted from the patients' charts and clinical notes.**

|  | Value |  |  |
| --- | --- | --- | --- |
| Chart Number | 82 | 93 | 104 |
| Date | 2019-10-18 | 2019-06-17 | 2019-06-17 |
| Systolic blood pressure | 132 | 144 | 134 |
| Diastolic blood pressure | 78 | 88 | 86 |
| Stage | III | IV | III |
| Grade | A | C | B |
| Heart rate | 72 | 68 | 82 |
| Stain | Medium | Heavy | Heavy |
| Plaque | Heavy | Heavy | Medium |
| Calculus | Heavy | Heavy | Medium |
| Bone loss | Generalized | Generalized | Generalized |
| Tooth mobility | Yes | Yes | No |
| Allergy | No | No | No |
| History of previous surgery | No | Yes | No |
| Diabetes | No | Yes | No |
| Smoking | No | Yes | No |
| Pocket score | Generalized | Generalized | Generalized |
| CAL score | Generalized | Generalized | Generalized |
| Count of teeth with bleeding | 12 | 23 | 14 |
| Count of teeth with plaque | 22 | 17 | 19 |

**Table 6. Confusion matrix evaluating the performance of the NN model in classifying the stage and grade of the periodontal disease using the contributing factors extracted from patients' charts by Regex and MLP.**

| Stages/grades | Precision | Recall | F-Score |
|---|---|---|---|
| I | 0.00 | 0.00 | 0.00 |
| II | 0.00 | 0.00 | 0.00 |
| III | 0.69 | 0.99 | 0.81 |
| IV | 0.00 | 0.00 | 0.00 |
| A | 0.00 | 0.00 | 0.00 |
| B | 0.72 | 0.54 | 0.61 |
| C | 0.21 | 0.41 | 0.28 |

grade A = 2, grade B = 21, and grade C = 9). Our proposed BERT model was able to predict the new unseen patients' stage and grade with 66% and 72% accuracy, respectively. Although the accuracy of the BERT model for the new patients' stages is high, the performance of the model for predicting the grade of the patients' is higher than the stage similar to the MLP model (59.4% vs. 62.5%).

During the evaluation, we closely analyzed the misclassified cases to identify patterns or common features leading to incorrect predictions. It was observed that a significant portion of misclassified cases belonged to the less prevalent stages and grades, indicating potential data imbalance issues. This analysis highlights the need for enhancing data augmentation techniques to better represent underrepresented classes in the dataset. According to Fig 2, the RegEx/MLP model generally performs better in reducing misclassifications, particularly for stage III cases, where it correctly identifies 17 out of 21 instances compared to BERT's 14.

**Table 7. Performance comparison between the BERT and MLP models on the new unseen dataset.**

| Stages | Precision | Recall | F1-Score |
|---|---|---|---|
| I | | | |
| RegEx | 0 | 0 | 0 |
| BERT | 1 | 1 | 1 |
| II | | | |
| RegEx | 0.22 | 0.5 | 0.30 |
| BERT | 0.25 | 0.25 | 0.25 |
| III | | | |
| RegEx | 0.74 | 0.81 | 0.77 |
| BERT | 0.78 | 0.67 | 0.72 |
| IV | | | |
| RegEx | 0 | 0 | 0 |
| BERT | 0.5 | 0.83 | 0.62 |
| Grades | | | |
| A | | | |
| RegEx | 0 | 0 | 0 |
| BERT | 0 | 0 | 0 |
| B | | | |
| RegEx | 0.75 | 0.86 | 0.8 |
| BERT | 0.78 | 0.86 | 0.82 |
| C | | | |
| RegEx | 0.33 | 0.22 | 0.27 |
| BERT | 0.62 | 0.56 | 0.59 |

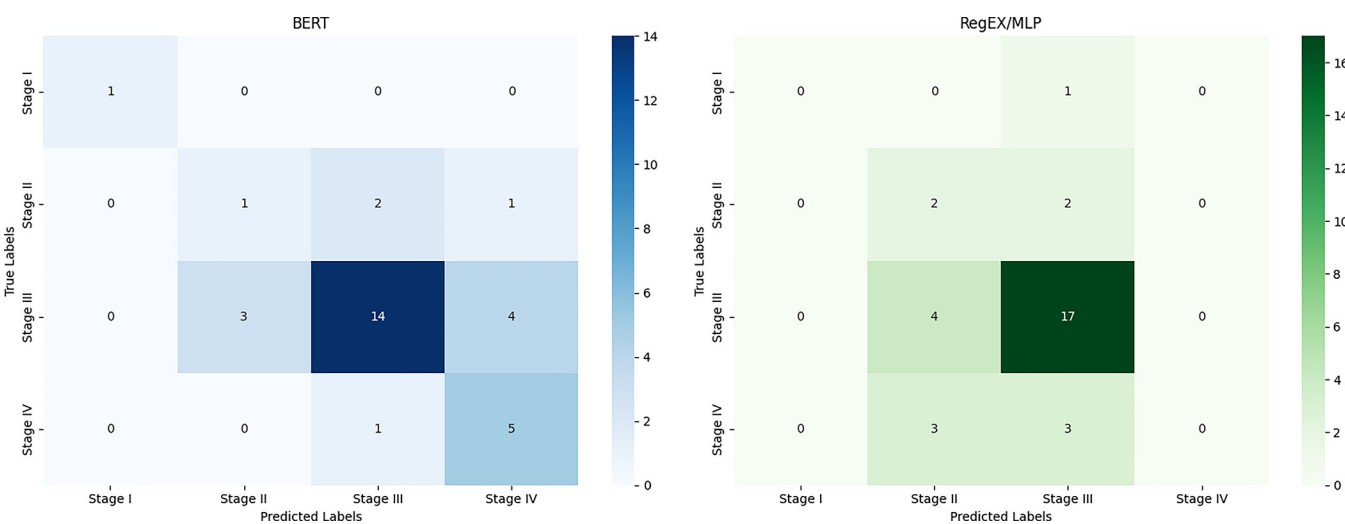

**Fig 2. Confusion matrices for stage classification using BERT and RegEx/MLP models.**

However, the BERT model exhibits slightly fewer misclassifications overall, especially for Stage IV cases, and handles stage I classifications well, with minimal confusion, indicating higher reliability in identifying early-stage cases (Fig 2).

The comparison between the BERT and RegEx/MLP models in the context of grade classification on the new unseen data, as illustrated in Fig 3, shows that both models perform similarly in correctly identifying grade B cases, with each model correctly classifying 18 instances. However, the BERT model exhibits a slightly better performance in reducing misclassifications for grade C cases, where it correctly classifies 5 out of 9 instances, compared to the RegEx/MLP model's 2. Both models show difficulty in correctly identifying grade A cases, with BERT misclassifying both instances as grade B and RegEx/MLP misclassifying them as grade C (Fig 3).

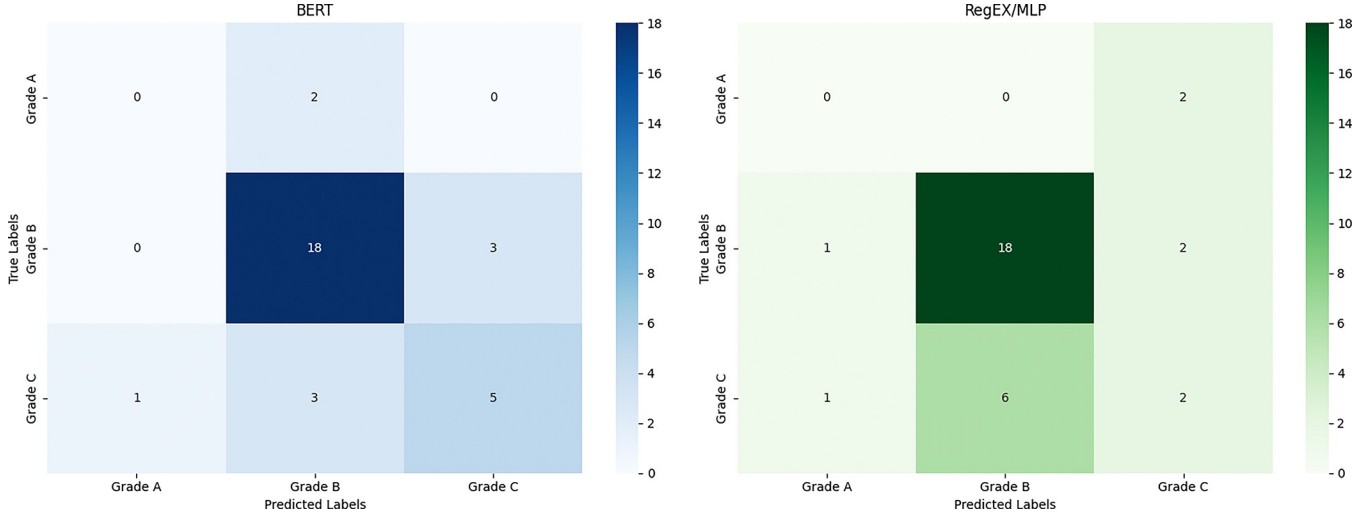

**Fig 3. Confusion matrices for grade classification using BERT and RegEx/MLP models.**

## Discussion

The increasing popularity of computerized dental records provides the opportunity to utilize AI-based technologies such as ML and DL models to improve patient care; however, a recent review has reported dentistry's clinical integration of such techniques has lagged [41]. To the authors' knowledge, there have only been a few studies that have investigated the use of ML and DL techniques in the diagnosis of periodontitis. Most of these studies have focused on analysing panoramic radiographs/and or clinical examinations or the biomarkers/bacteria extracted from the saliva [2], which is costly and time-consuming. Our study compared two approaches for automatically extracting patient data from textual notes and periodontal charts using ML techniques and demonstrated their ability to classify the stage and grade of the periodontitis.

In the first phase, we applied the BERT model to predict the stage and grade of the periodontal disease using the patients' textual notes. Our results demonstrated that the BERT model outperformed other applied methods in terms of accuracy in classifying the correct stage and grade of the patients' periodontal disease. To date, no previous studies have used patients' dental records collected as text files for analysis via BERT algorithms. In medicine, Haulcy and Glass compared the performance of five classifiers, as well as convolutional neural networks and long- and short-term memory networks on the classification of Alzheimer's disease using audio features and text features [42]. They reported that the top-performing classification models were the support vector machine and random forest classifiers trained on BERT embeddings, which both achieved an accuracy of 85.4% on the test set. Recently, new domain-specific BERT models pretrained on large-scale biomedical (BioBERT) and medical dictionaries (Med-BERT) have been introduced. These models have shown promising results in extracting valuable information from biomedical and medical literature, respectively, and outperformed the state-of-the-art methods on medical and biomedical tasks [43,44]. However, we used a base architecture in the study's BERT model, as the dental patients' notes mostly include a detailed document of the dental history, physical examination, diagnosis, and treatment planning, which is different from the biomedical or medical records. Moreover, as the study is the very first application of BERT model on patients' dental notes, we decided to use a more generic architecture.

In the second phase, we utilized baseline feature engineering (RegEx) and MLP models to prepare the patient data and classify the stage and grade of the disease. Integrating these AI models into clinical workflows could enable real-time analysis of patient data directly from digital records, providing immediate diagnostic support for clinicians. This integration would streamline the diagnostic process, reduce the need for additional tests, and potentially lower costs. However, challenges such as data standardization, interoperability among EDR systems, and clinician training must be addressed to ensure seamless integration. Risks include the possibility of over-reliance on AI predictions, which might lead to misdiagnosis if the model's limitations are not well understood or if data quality issues are not adequately managed.

Currently digital dental and medical records are gaining popularity for storing patient information. Data in electronic medical/dental records can be divided into three kinds: structured data, semi-structured data, and unstructured data. Unstructured text is defined as narrative data, consisting of clinical notes, discharge records, and radiology reports. Unstructured texts store a lot of valuable information however, common structural frameworks are usually lacking, and many errors including improper grammatical use, spelling errors, local dialects, and semantic ambiguities exist, which result in the complexity of data processing and analysis. Retrieval and application of such valuable information is possible through the application of text mining methods [18]. We used MLP to build our network, as its implementation is easy.

MLP was also chosen as it is capable of providing high-quality models, while keeping the training time relatively low [45]. The proposed ML method in phase 1 was able to extract relevant information and classify the stage and grade of the periodontitis with approximately 70% accuracy, which indicates that this model is well-suited to automatically classify the periodontitis stage and grade using patient chart and clinical data.

In comparison to traditional clinical methods and other published approaches for periodontal staging and grading, our BERT model demonstrated promising results by just utilizing clinical notes without any imaging data. In contrast to Ertas et al [39], our study focused on using textual data from patient notes, which is often more readily available in clinical settings compared to comprehensive radiographic data. Our BERT model achieved an accuracy of 72% for grading and 66% for staging, which aligns closely with the performance levels of periodontal specialists observed by Oh et al., who reported accuracies of 71.33% for staging and 64% for grading among clinicians with periodontal backgrounds [46] This suggests that the use of NLP techniques like BERT can potentially bridge the gap between non-specialists and specialists in clinical diagnosis.

Additionally, in a recent study by Tastan Eroglu et al. [47], the ChatGPT model was used to classify periodontitis based on textual inputs, with staging and grading accuracies of 59.5% and 50.5%, respectively. These values are lower compared to our BERT and even RegEx models' performance, highlighting BERT's superiority in understanding clinical text for periodontitis classification. This difference may be attributed to BERT's fine-tuning capabilities on domain-specific data, which enables it to capture more nuanced clinical features compared to general-purpose models like ChatGPT.

These comparisons indicate that while the BERT model may not yet match the highest accuracies achieved with image-based methods for staging, its performance is superior for text-based classification tasks. This demonstrates its potential utility in settings where detailed radiographic data may not be available, making it a valuable tool for enhancing decision support in clinical practice. However, future studies should aim to integrate multimodal data—combining both textual and radiographic information—to leverage the strengths of each data type and further improve classification accuracy.

The study by Oh et al. [46] revealed that dental practitioners with periodontal backgrounds had higher accuracy in classifying periodontitis stages (71.33%) compared to grades (64%), with non-periodontal practitioners showing a similar trend but with lower accuracy (61.67% for stages and 49.33% for grades). In contrast, our BERT model demonstrated higher accuracy in predicting grades (72%) than stages (66%) for new patients, with these results being closer to the accuracy levels achieved by periodontal specialists. Additionally, the BERT model's classification accuracy for both stage and grade surpasses that of non-periodontal practitioners, while the MLP model, with an accuracy of 62.5% for grades and 59.4% for stages, also outperforms non-periodontal clinicians. These comparisons suggest that our BERT model aligns more closely with specialist performance but both approaches exceed the accuracy of non-specialists in classifying the periodontitis stage and grade, indicating their strong potential for effective clinical application in the disease classification.

As the current study aimed to examine several different health-related medical and dental possible contributing factors using patient data and periodontal charts, ML methods were best suited to address the complexity and multi-dimensional nature of each due to their known ability to detect complex relationships and classify the outcome accurately [41]. ML models outperform the traditional statistical methods as they can consider a broader range of features without strict, predetermined predictor and outcome parameters and classify the output with impressive accuracy [48].

Text mining techniques, which allow for the extraction of high-quality information from large-scale unstructured text data can acquire implicit knowledge that is hidden in the unstructured text through extracting the predefined information and new knowledge from the unstructured texts [49]. In the first phase, we applied RegEx to complete the text mining procedure. In a study conducted by Zhen et al. to compare two text-mining methods (RegEx and Naïve Bayes Classifier) for analysing published full articles in terms of their adoption of standards in radiation therapy [35], they found that classifications and overall usage trends reported by the RegEx-based method are comparable to those of the domain expert.

Despite its promising results in medical research, utilization of DL and NN in dental research generally and in the periodontal field specifically is relatively scarce. A recent review by Ossowska et al. [50] has shown that new technologies are developing very quickly in the field of dentistry and AI is spreading into periodontology with the greatest focus on evaluating periodontal bone loss, peri-implant bone loss, and predicting the development of periodontitis due to the psychological features.

The potential utility of applying ML and DL models to automatically classify the stage and grade of periodontitis using patients' notes and clinical charts have been found in the current study. The two methods presented, phase one relying on a combination of chart and text mining, and phase two relying on intelligent text mining only, both yielded promising results. However, these findings must be interpreted with caution according to the preliminary nature of this research as it is a novel method applied on unstructured patient notes and charts recruited from a university periodontal clinic. Therefore, the text mining and DL techniques (MLP and BERT) used in the present study require validation to confirm their efficiency on different types of medical/dental records from other centers.

The model demonstrated higher accuracy in classifying stage III and grade B periodontitis, which can be attributed to the imbalanced nature of the input data, with a greater number of patients falling into these categories compared to stages I or II and grade A. This imbalance is likely due to the increased number of referrals for patients with more severe periodontal conditions. Despite the relatively small sample size of 309 patients—a recognized limitation for a ML study—our findings indicate a relatively high accuracy in classifying the stage and grade of periodontitis. It is crucial to acknowledge that this class imbalance, with a predominance of patients in stage III and grade B, inherently influenced the model's performance, partly due to the higher referral rates for severe cases. The model's ability to achieve higher accuracy in these classes, even with a limited sample size, underscores its robustness in handling imbalanced data. To address potential overfitting, we incorporated regularization techniques, including dropout layers and early stopping, in the model training process. Despite these measures, a slight tendency towards overfitting was observed, particularly in the later stages of training.

Future studies should also analyze larger and more balanced datasets using these ML/DL techniques to provide additional validation. Specifically, exploring techniques such as synthetic data generation and data augmentation could be valuable in overcoming the challenge of class imbalance and enhancing model performance. Moreover, evaluating the integration of other machine learning approaches, like ensemble methods, may further improve classification accuracy. However, the ethical concerns surrounding AI in medical diagnosis, particularly issues related to bias, transparency, and patient privacy, have led to restrictions on access to large, diverse datasets in healthcare [40,51]. This limited access exacerbates the challenge of developing unbiased and accurate AI models, as the scarcity of data hinders the ability to train AI systems effectively. Without comprehensive datasets, AI models risk perpetuating existing disparities in healthcare, as they may not adequately represent all patient populations [52].

Addressing these ethical challenges is essential to balance the need for robust AI development with the protection of patient rights and the promotion of fair, transparent medical practices.

In spite of access to few numbers of data in our study, and the novelty of our proposed method, our findings showed a relatively high accuracy in correct classification of the stage and grade of the periodontitis, which suggests that our proposed method was successful in extracting the relevant information from patient notes and classifying the stage and grade of the periodontal disease.

One limitation of the current study is that we did not conduct a manual review of the 309 cases using both methods among the authors. The rational for this was first, since each model in the present study utilized a randomly sampled 70% of cases for training, separately for BERT and the feature method, it is likely that the models would perform well in such a comparison. However, this would introduce bias, as the validation data (30%) was not used in the training process. Second, the two models used different validation data, and there were very few overlaps. Therefore, it is not straightforward to compare the outcomes directly.

Our new proposed models are the preliminary applications of NLP and DL techniques to assign the stage and grade of the periodontal disease using patients' notes compared to the previous models of utilizing radiographic images and salivary samples. This model can identify the stage and grade of the periodontitis without the need for a comprehensive clinical and radiographic examination, which are routinely required for disease diagnosis. Future research should focus on longitudinal studies to assess the predictive value of these models over time, along with a detailed comparison of different NLP and DL architectures to identify the most efficient approaches for clinical implementation. We hope this will increase enrollment in clinical trials of new therapies, and improve patient outcomes by enabling periodontists to diagnose periodontal disease using the information collected in patients' notes in an accurate and timely manner. However, these models are not intended to replace the clinical judgment of experienced practitioners but to serve as a valuable support tool, particularly for general practitioners or those with limited experience in diagnosing periodontitis. These techniques can be also applied to other health-care providers in need of using patients' information as large amounts of unstructured texts to classify stages of a disease. It is recommended that future studies compare various BERT models and other NLP techniques to identify the most effective ones for extracting information from EDR to improve classification accuracy and clinical applicability.

## Author Contributions

**Conceptualization:** Nazila Ameli, Monica Gibson, Hollis Lai.

**Data curation:** Nazila Ameli.

**Formal analysis:** Nazila Ameli, Tahereh Firoozi, Hollis Lai.

**Funding acquisition:** Nazila Ameli, Hollis Lai.

**Investigation:** Nazila Ameli.

**Methodology:** Nazila Ameli, Tahereh Firoozi.

**Project administration:** Hollis Lai.

**Resources:** Monica Gibson.

**Supervision:** Monica Gibson, Hollis Lai.

**Validation:** Nazila Ameli, Tahereh Firoozi.

**Writing – original draft:** Nazila Ameli.

**Writing – review & editing:** Tahereh Firoozi, Monica Gibson, Hollis Lai.

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
