## [Decision Letter · Decision Letter 0]

23 Jul 2024

PDIG-D-24-00165

Classification of periodontitis stage and grade using natural language processing techniques

PLOS Digital Health

Dear Dr. Ameli,

Thank you for submitting your manuscript to PLOS Digital Health. After careful consideration, we feel that it has merit but does not fully meet PLOS Digital Health's publication criteria as it currently stands. Therefore, we invite you to submit a revised version of the manuscript that addresses the points raised during the review process.

Please submit your revised manuscript within 60 days Sep 21 2024 11:59PM. If you will need more time than this to complete your revisions, please reply to this message or contact the journal office at digitalhealth@plos.org. Please include the following items when submitting your revised manuscript:

We look forward to receiving your revised manuscript.

Kind regards,

Akhilanand Chaurasia

Section Editor

PLOS Digital Health

Journal Requirements:

Additional Editor Comments (if provided):

The study on periodontitis classification using BERT and RegEx/MLP approaches has several limitations that impact its reliability and generalizability. The small sample size from a single clinic, data imbalance, and lack of comparison with other state-of-the-art methods raise concerns about the model's performance and applicability. Additionally, the paper fails to address important aspects such as model interpretability, overfitting prevention, and ethical considerations in medical AI applications.

The technical details provided in the paper are insufficient for a thorough evaluation. Key information about the RegEx implementation, BERT model specifications, and LSTM integration is missing. The study also lacks crucial data distributions and explanations for performance discrepancies across different stages and grades. Furthermore, the authors did not consider alternative approaches such as using specialized biomedical language models or exploring zero-shot learning with large language models, which could potentially yield better results.

Reviewers' comments:

Reviewer's Responses to Questions

**Comments to the Author**

1. Does this manuscript meet PLOS Digital Health’s publication criteria? Is the manuscript technically sound, and do the data support the conclusions? The manuscript must describe methodologically and ethically rigorous research with conclusions that are appropriately drawn based on the data presented.

Reviewer #1: Yes

Reviewer #2: No

2. Has the statistical analysis been performed appropriately and rigorously?

Reviewer #1: Yes

Reviewer #2: No

3. Have the authors made all data underlying the findings in their manuscript fully available (please refer to the Data Availability Statement at the start of the manuscript PDF file)?

Reviewer #1: Yes

Reviewer #2: No

4. Is the manuscript presented in an intelligible fashion and written in standard English?

Reviewer #1: Yes

Reviewer #2: Yes

5. Review Comments to the Author

Reviewer #1: This study explores the use of NLP and machine learning techniques to classify the stage and grade of periodontitis using patient data from dental records. The researchers developed and compared two approaches: one using a BERT model, and another using a combination of RegEx for text mining and a MLP neural network. They analyzed 309 anonymized patient periodontal charts and corresponding clinician's notes from a university periodontal clinic. The BERT model achieved higher accuracy in predicting both the stage (77%) and grade (75%) of periodontitis compared to the RegEx and MLP approach, which achieved about 70% accuracy for both.

Comments

The study used only 309 patient records from a single university clinic. This small, potentially homogeneous sample may not be representative of the broader population, limiting generalizability.

The authors noted an imbalance in the input data, with more patients in stage III and grade B. This can lead to biased model performance.

While the study compared BERT with a RegEx/MLP approach, it didn't explore other state-of-the-art NLP or machine learning methods.

The study doesn't provide insight into which features or text patterns are most important for classification. Incorporate model interpretability techniques like SHAP values or LIME.

The paper doesn't provide a detailed analysis of misclassified cases.

While the models show good statistical performance, there's no indication of how well they align with expert clinical judgment.

The study doesn't mention techniques used to prevent overfitting, especially given the small dataset.

The paper doesn't address potential ethical concerns of using AI for medical diagnosis.

The paper lacks some important details about model architecture, hyperparameters, and training procedures.

Reviewer #2: Nazila et al. developed a BERT model to automatically classify periodontitis grade and stage. The model was claimed to have 77% and 75% accuracy on classification, which is better than a MLP baseline. However, due to the missing of detailed description of both model and data, it is very difficult to evaluate the validity of the claims in the paper. Please see the comments below:

How difficult it is to determine stage and grade manually? Does the EDR notes usually contain the stage and grade information already? 

The technical details in the paper were missing, which makes is very difficult to evalute the paper. For example, what is the exact regular expression used to extract information? Which BERT model was used as base model for fine tuning? How were LSTM combined with BERT?

Why not use BioBERT or MedBERT as base model?

Can RegEx accurately extract the information from notes and charts? It is not clear to me why regular expression can be used to extract information from chart? What is the main bottleneck of the RegEx features based MLP? RegEx part or MLP part?

Can the authors show distributions of import statistics of the dataset? e.g. how many datapoint are in stage I, II, III, and grade A, B, C?

Is there a reason why only stage III and grade B have good performance?

Before branding the proposed the model as diagnostic tool, there are so many issues that needs to be solved. For instance, would the model be used only after being manually diagnosis of periodnotitis? Is there any negative data being used in the study (e.g. data from other close disease or healthy patient)?

Does the authors consider using existing large language models (LLM) for zero or few shot learning? LLMs are a good baseline to compare with the propose model. For instance, with proper prompt engineering for Llama3 or GPT4 by including the standard diagnostic standard as system prompt and the EDR notes as user prompt. In addition, even for the task of extracting information from EDR, LLM should be better than RegEx as RegEx does not understand sematic information.

Can the authors comment on how robust the model is? What will happen if the model was applied to notes that is written by another dentists?

6. PLOS authors have the option to publish the peer review history of their article (what does this mean?). If published, this will include your full peer review and any attached files.

**Do you want your identity to be public for this peer review?** For information about this choice, including consent withdrawal, please see our Privacy Policy.

Reviewer #1: No

Reviewer #2: No

---

## [Decision Letter · Decision Letter 1]

1 Oct 2024

PDIG-D-24-00165R1

Classification of periodontitis stage and grade using natural language processing techniques

PLOS Digital Health

Dear Dr. Ameli,

Thank you for submitting your manuscript to PLOS Digital Health. After careful consideration, we feel that it has merit but does not fully meet PLOS Digital Health's publication criteria as it currently stands. Therefore, we invite you to submit a revised version of the manuscript that addresses the points raised during the review process.

Please submit your revised manuscript within 60 days Nov 30 2024 11:59PM. If you will need more time than this to complete your revisions, please reply to this message or contact the journal office at digitalhealth@plos.org. Please include the following items when submitting your revised manuscript:

We look forward to receiving your revised manuscript.

Kind regards,

Wisit Cheungpasitporn, MD

Academic Editor

PLOS Digital Health

Additional Editor Comments (if provided):

The reviewers suggest additional several areas for improvement and further exploration. They recommend incorporating model interpretability techniques, providing a detailed analysis of misclassified cases, and addressing potential overfitting issues. Additionally, they advise exploring the alignment between model performance and expert clinical judgment, addressing ethical concerns related to AI in medical diagnosis, and including more details about model architecture, hyperparameters, and training procedures. Overall, while the study shows promise, the reviewers suggest that addressing these concerns would strengthen the research and its potential impact in the field.

Reviewers' comments:

Reviewer's Responses to Questions

**Comments to the Author**

1. If the authors have adequately addressed your comments raised in a previous round of review and you feel that this manuscript is now acceptable for publication, you may indicate that here to bypass the “Comments to the Author” section, enter your conflict of interest statement in the “Confidential to Editor” section, and submit your "Accept" recommendation.

Reviewer #1: All comments have been addressed

Reviewer #2: (No Response)

2. Does this manuscript meet PLOS Digital Health’s publication criteria? Is the manuscript technically sound, and do the data support the conclusions? The manuscript must describe methodologically and ethically rigorous research with conclusions that are appropriately drawn based on the data presented.

Reviewer #1: Yes

Reviewer #2: Partly

3. Has the statistical analysis been performed appropriately and rigorously?

Reviewer #1: Yes

Reviewer #2: No

4. Have the authors made all data underlying the findings in their manuscript fully available (please refer to the Data Availability Statement at the start of the manuscript PDF file)?

Reviewer #1: Yes

Reviewer #2: No

5. Is the manuscript presented in an intelligible fashion and written in standard English?

Reviewer #1: Yes

Reviewer #2: Yes

6. Review Comments to the Author

Reviewer #1: Overall, the manuscript shows notable improvement in terms of technical detail and transparency. However, there are still some important considerations around clinical applicability and broader context that could be addressed to strengthen the paper further. The core contribution - applying BERT to periodontal staging and grading - remains novel and potentially impactful, but the limitations of the current study should be clearly communicated. 

Major comments

While acknowledged as a limitation, the relatively small sample size (309 patients) is still a significant concern for a machine learning study, especially given the class imbalance.

While LIME analysis is mentioned, there could be more in-depth discussion of what features the models are using to make predictions and how this aligns with clinical understanding.

The discussion could be expanded to more clearly articulate how these models could be integrated into clinical workflows and what potential benefits/risks this might entail.

While the study compares BERT to RegEx/MLP, it would be valuable to see how these methods compare to current clinical practice or other published approaches for periodontal staging and grading.

The conclusion could be strengthened by more specific suggestions for future research to address the current limitations.

Reviewer #2: (No Response)

7. PLOS authors have the option to publish the peer review history of their article (what does this mean?). If published, this will include your full peer review and any attached files.

**Do you want your identity to be public for this peer review?** For information about this choice, including consent withdrawal, please see our Privacy Policy. 

Reviewer #1: No

Reviewer #2: No

---

## [Decision Letter · Decision Letter 2]

6 Nov 2024

Classification of periodontitis stage and grade using natural language processing techniques

PDIG-D-24-00165R2

Dear Dr. Ameli,

We are pleased to inform you that your manuscript 'Classification of periodontitis stage and grade using natural language processing techniques' has been provisionally accepted for publication in PLOS Digital Health.

Best regards,

Wisit Cheungpasitporn, MD

Academic Editor

PLOS Digital Health

**Additional Editor Comments (if provided):**

It is evident that all concerns raised have been adequately addressed. The manuscript is well-written and demonstrates substantial improvement. I have no additional comments and recommend acceptance for publication.

**Reviewer Comments (if any, and for reference):**

Reviewer's Responses to Questions

**Comments to the Author**

1. If the authors have adequately addressed your comments raised in a previous round of review and you feel that this manuscript is now acceptable for publication, you may indicate that here to bypass the “Comments to the Author” section, enter your conflict of interest statement in the “Confidential to Editor” section, and submit your "Accept" recommendation.

Reviewer #1: All comments have been addressed

2. Does this manuscript meet PLOS Digital Health’s publication criteria? Is the manuscript technically sound, and do the data support the conclusions? The manuscript must describe methodologically and ethically rigorous research with conclusions that are appropriately drawn based on the data presented.

Reviewer #1: Yes

3. Has the statistical analysis been performed appropriately and rigorously?

Reviewer #1: Yes

4. Have the authors made all data underlying the findings in their manuscript fully available (please refer to the Data Availability Statement at the start of the manuscript PDF file)?

Reviewer #1: Yes

5. Is the manuscript presented in an intelligible fashion and written in standard English?

Reviewer #1: Yes

6. Review Comments to the Author

Reviewer #1: After careful consideration of the revised manuscript and the authors' point-by-point responses, I conclude that the raised issues have been adequately addressed. The revisions have sufficiently improved the manuscript, and I have no additional comments to put forward. I endorse the acceptance of this manuscript in its present state.

7. PLOS authors have the option to publish the peer review history of their article (what does this mean?). If published, this will include your full peer review and any attached files.

**Do you want your identity to be public for this peer review?** For information about this choice, including consent withdrawal, please see our Privacy Policy.

Reviewer #1: No
